# Improved Yield and Electrical Properties of Poly(Lactic Acid)/Carbon Nanotube Composites by Shear and Anneal

**DOI:** 10.3390/ma16114012

**Published:** 2023-05-27

**Authors:** Dashan Mi, Zhongguo Zhao, Haiqing Bai

**Affiliations:** 1School of Mechanical Engineering, Shaanxi University of Technology, Hanzhong 723001, China; 2School of Materials Science and Engineering, Shaanxi University of Technology, Hanzhong 723001, China; zhaozhongguo@snut.edu.cn

**Keywords:** polylactic acid (PLA), carbon nanotube, mechanical properties, conductivity

## Abstract

Shear and thermal processing can greatly influence nanoparticles’ orientation and dispersion, affecting the nanocomposites’ conductivity and mechanical properties. The synergistic effects of shear flow and Carbon nanotubes (CNTs) nucleating ability on the crystallization mechanisms have been proven. In this study, Polylactic acid/Carbon nanotubes (PLA/CNTs) nanocomposites were produced by three different molding methods: compression molding (CM), conventional injection molding (IM), and interval injection molding (IntM). Solid annealing at 80 °C for 4 h and pre-melt annealing at 120 °C for 3 h was applied to research the CNTs’ nucleation effect and the crystallized volume exclusion effect on the electrical conductivity and mechanical properties. The volume exclusion effect only significantly impacts the oriented CNTs, causing the conductivity along the transverse direction to rise by about seven orders of magnitude. In addition, the tensile modulus of the nanocomposites decreases with the increased crystallinity, while the tensile strength and modulus decrease.

## 1. Introduction

Polylactic acid (PLA) attracts attention due to its bio-friendly advantages. It is polymerized from bio-sources such as sugar cane and corn starch, and is biocompatible and biodegradable [1,2,3,4]. In addition, PLA has outstanding strength and hardness [5], which makes it the most promising bio-based polymer to replace traditional petrol-based polymers such as polypropylene and polyethylene. However, PLA exhibits two significant drawbacks: slow crystallization kinetics and low thermal stability under conventional processing conditions [6]. Considering the slow crystallization kinetics, the final part is almost amorphous. Improving its crystallization kinetics becomes crucial to enhancing the final properties (strength, dimensional stability, and flexural modulus) and extend its uses to a wild field.

Carbon nanotubes (CNTs) have excellent potential to offer extraordinary mechanical properties, antibacterial efficiency, and good electrical and thermal conductivity [7,8,9]. The CNTs aspect ratio, filler-filler interaction, polymer-filler interaction, and other processing parameters (shear rate, mixing time, and temperature) are among the factors that contribute to the final nanocomposite’s properties [10,11]. When PLA is reinforced with nanoparticles, the desired performance is created for advanced applications [12]. However, due to their poor interaction, inferior interfacial adhesion and the poor dispersion of CNTs always occur when compositing polymers with CNTs [13].

High shear can improve the mechanical properties and CNTs dispersion, while adding CNTs may further enhance shear and promote thick shear layer formation [14]. For example, Mei et al. [15] suggest that shear can disperse and align the CNTs along the flow direction in micro injection molding. Bai et al. use extensional flow to induce conductive nanohybrid shish in PLA nanocomposites and significantly improve the strength and toughness. In addition, there are currently two theories regarding the relationship between CNTs and shear: independent and synergy effects theories. The former believes that the nucleating agent and the shear flow contribute independently to the crystallization process [16]. It is known as the additive effect: the flow promotes homogeneous nucleation, and the density of nuclei depends on the shearing conditions, while the nuclei contribute separately with a fixed number of CNTs. Another theory involves the synergy effects between the particles and shear flow in terms of crystallization; this synergy can be observed as supplementary nucleation induced by their interaction. The synergy effects seemed more suitable for PLA/CNTs nanocomposites with regard to explaining the increase of the crystallinity [17].

Shear, crystallization, and conductive networks have a more complex relationship. For example, Quan et al. [18] showed that the formation of stereocomplex crystals in a blend of poly(L-lactic acid) (PLLA) and poly(D-lactic acid) (PDLA) produced a volume exclusion effect. The CNTs had been diffused out of the crystals, significantly decreasing the percolation threshold [18]. However, in other cases, several research works have suggested that the crystal nucleation ability of CNTs for semicrystalline polymers results in the wrapping of CNTs by a thick layer of polymer crystals [19,20]. This indicates that CNTs will cause the crystals to nucleate on their surface and disrupt the conductive networks by encapsulating and isolating the individual CNTs. Therefore, there exist two opposite mechanisms to describe the effect of crystals on the electrical conductivity in PLA/CNTs nanocomposites. One is that crystal growth can diffuse the CNTs into the amorphous regions, causing a volume exclusion effect and promoting conductive network formation. Another depends on the crystal nucleation ability of CNTs, which can wrap the CNTs and destroy the conductive network.

This work will explore the aforementioned two mechanisms in terms of the conductive network under different shear conditions. The annealing method promotes the growth of PLA crystals, and then helps find the crystal’s effect on the conductive network. Annealing on PLA was reported to be an efficient treatment to increase modulus and tensile strength, which is the cause of the increased crystallinity. Various time and temperature conditions were applied for the annealing of PLA, and the crystallinity and melting temperature of PLA increased with increasing the annealing temperature in the range of 100–140 °C [21,22,23]. The CNTs loading, in conjunction with annealing, may significantly affect the melting behavior, the glass transition temperature, crystallinity, and the mechanical properties of PLA. In addition, for semi-crystalline polymers, shear combined with annealing has the potential to synergistically enhance mechanical strength [24,25,26].

As far as we know, there is little research on the influence of crystallization on the formation of conductive networks under different CNTs oriented degrees. In the present work, three different processing methods were used to provide three kinds of shear strength, from weak to strong, and explored the influence of shear and annealing on the changes of orientation, crystallinity, and CNTs distribution of PLA/CNT nanocomposites. Moreover, the CNTs nucleation effect and crystallized volume exclusion effect are discussed for various oriented CNTs, and we then explore their ultimate impact on the conductive network and mechanical properties.

## 2. Experimental Section

### 2.1. Materials and Sample Preparation

Commercial grade polylactic acid (PLA, trade name 8052D) was purchased from Nature Works LLC, Minneapolis, MN, USA. PLA samples were dried for 5 h at 80 °C. CNTs were purchased from Suzhou Tanfeng Graphene Technology Co., Suzhou, China. CNTs are multi-walled with an average diameter of 50 nm, an average length of 20 μm, and a middle surface area of 300 m^2^/g (according to the supplier), with a carbon purity of 90%.

Figure 1 shows polymer composites with filler contents of 6 wt.% that were blended by a micro-conical double-screw extruder (SJZS-10A, Wuhan Ruiming Co., Wuhan, China) at a barrel temperature of 200 °C. Three molding methods were then used to provide different shear stress, such as Compression molding (CM), Injection molding (IM), and Interval injection molding (IntM).

CM provides a low shear condition (Figure 1a). Plates were compression-molded with 1.2 mm thickness at 200 °C. The plate was subsequently cut into a dumbbell shape.

IM provides a moderate shear condition (Figure 1b). we melt-filled the mold chamber under constant pressure. Dumbbell-type samples of 2 mm were prepared by a micro conventional injection molding (SJZS-10A, Wuhan Ruiming Co., Wuhan, China) with a 50 °C mold temperature and a 200 °C barrel temperature.

IntM uses the same equipment and temperature as IM. IntM can provide a high shear condition, as shown in Figure 1c. The melt filled the mold from two flows. A gradient pressure setting can achieve this; the initial injection pressure is too low to fill the cavity. One second after the short shot, the filling will be completed by a second flow with a higher pressure. It is worth mentioning that most commercial injection equipment can implement this pressure setting to magnify the shear stress.

Two temperature conditions were applied for the annealing of the molded samples: (i) solid annealing at 80 °C for 4 h (above the glass transition); and (ii) pre-melt annealing at 120 °C for 3 h (above the cold crystallization and before melting).

All samples were labeled according to CNTs content, molding method, and annealed temperature. For example, 6IntM80 means that the CNTs content is 6wt.%, molded by IntM, and annealed at 80 °C. An example of the label method is also shown in Table 1.

### 2.2. Sample Testing

#### 2.2.1. Measurement of Mechanical Properties

The tensile test was conducted at room temperature (26 °C) on an electro-universal testing machine (GOTECH-20KN, GOTECH Testing Machines CO., Dongguan, China) with a 2 mm/min cross-head speed. The values of the properties were calculated as the averages of five samples.

#### 2.2.2. Measurement of Electrical Conductivity

The electrical conductivity was measured by a Keithley 6487 source meter. Both ends of the specimens were coated with conductive silver paint. The electrical conductivity (σ) was then calculated according to the following:σ = L/RA(1)

L is the distance between the electrodes, R is the measured resistance, and A is the cross-sectional area. For IM and IntM samples, the σ was measured along the flow and transverse directions. The values of the σ were calculated as the averages of the four samples.

#### 2.2.3. X-ray Measurements

The synchrotron 2D-WAXD experiment was carried out on HomeLab (Rigaku, Tokyo, Japan). Specimens cut from the middle of the bars were manufactured into the samples with a dimension of 8 × 2 × 2 mm. X-rays were imaged after penetrating the piece, so the entire low- and high-shear area information was included in the image. The dimension of the rectangle-shaped beam was 100 × 100 μm^2^, and the wavelength of light was 0.154 nm.

The crystallinity index (*X_c_*) was calculated using the following equation [27]:(2)$$X_{c}=\frac{A_{c}}{A_{c}+A_{a}}\times 100\%$$
where *A_c_* is the fitting intensity of the crystallization peaks, and *A_a_* is the appropriate intensity of the amorphous phase.

The orientation degree of PLA crystals was calculated using Herman’s orientation function based on the (200)/(110) α plane reflections at 2θ = 16.7° in PLA crystals [28]. In this method, the crystal orientation was characterized by the average orientation of the normal to the crystal plane concerning an external reference frame. Accordingly, the flow direction was taken as the reference direction. For a set of hkl planes, the average orientation, expressed as ${\left(\cos^{2}\varphi \right)}_{\mathrm{hkl}}$, was calculated mathematically by using the following equation:(3)$${\left(\cos^{2}\varphi \right)}_{\mathrm{hkl}}=\frac{\int_{0}^{\pi /2}I\left(\varphi \right)\cos^{2}\varphi \sin\varphi d\varphi }{\int_{0}^{\pi /2}I\left(\varphi \right)\sin\varphi d\varphi }$$
where *φ* is the azimuthal angle, and *I(φ)* is the scattered intensity along the angle *φ*. Herman’s orientation function, *f*, is defined as
(4)$$f=\frac{3{\left(\cos^{2}\varphi \right)}_{\mathrm{hkl}}-1}{2}$$
where *f* has a value of −0.5 with the normal of the reflection plane being perpendicular to the reference direction (*φ* = 90°), a value of 1 with the normal of the refection plane parallel being the reference direction (*φ* = 0°), and a value of 0 with the orientation being random.

#### 2.2.4. Optical Microscopy (OM)

Thin slices cut using a microtome were used for optical morphology observations. The observation zones were along the flow direction. ImageJ 1.48V software was used to calculate the dispersion of the CNTs.

#### 2.2.5. Scanning Electron Microscopy (SEM)

A scanning electron microscope (ZEISS Gemini 300, ZEISS Microscopy LLC, Jena, Germany) was used for the SEM observations. The specimens were gold-sputtered after being yield stretched.

#### 2.2.6. Differential Scanning Calorimetry (DSC)

The melting behavior of the samples was evaluated on a TA Discovery DSC 250 (TA Instruments Co., New Castle, DE, USA) in a nitrogen atmosphere. The percentage of crystallinity, *cc*, was determined by the following equation:(5)$$X_{c}=\frac{\Delta H_{m}-\Delta H_{cc}}{\Delta H_{100}}\times 100\%$$
where $\Delta H_{m}$ is the enthalpy of melting, $\Delta H_{cc}$ is the enthalpy of cold crystallization, and $\Delta H_{100}$ is the theoretical enthalpy of melting for a 100% crystalline polymer, which is 93 J g^−1^ for PLA [29].

## 3. Results and Discussion

### 3.1. Sample Structure

#### 3.1.1. Crystallite and Orientation

As shown in Figure 2a, without CNTs, 0CM obtained two complete circles, namely lattice plane (200)/(110) α and lattice plane (203) α. No clear diffraction rings can be observed in 0IM, indicating low crystallinity or only an imperfect crystal. Weak arcs of (200)/(110) can be observed in 0IntM, indicating that the strong shear of IntM can slightly promote the formation of oriented crystals in PLA. As shown in Figure 2d, with the addition of 6 wt.% of CNTs, 6IntM obtains clear diffraction images. The reflection of lattice plane (200)/(110) α located on the meridian is the strongest. In contrast, the reflection of (203) α locating on the quadrant is comparatively weak, and the absence of (015) α and (206) α indicates the oriented crystalline consists of imperfect crystal [5]. After solid annealing at 80 °C for 4 h and pre-melt annealing at 120 °C for 3 h, the diffraction ring became clearer, indicating an increase in crystallinity and an improvement in the crystal structure. Figure 2g,h show the azimuth angle and liner-WAXD curves, respectively. Without the addition of CNTs, it is hard to observe significant peaks in liner-WAXD curves of IM and IntM, which means that almost no crystal can be detected by WAXD. The peak intensity of azimuth angle and linear-WAXD curves gradually increases with the addition of CNTs and the annealing treatment.

Figure 3 shows the crystallinity and orientation of each sample. The orientation degree of 0CM is the lowest, and the orientation degree of 0IM rises to 0.48 due to the effect of the injection molding shear. The high shear provided by IntM further increases the orientation degree of 0IntM to 0.66. Furthermore, with the addition of CNTs, the orientation degree of 6IntM further jumps to 0.81. With the increased crystal orientation, the CNTs also get a chance to form at a one-dimensional orientation, which may be similar to that of a one-dimensionally oriented block copolymer [30]. Afterward, solid annealing at 80 °C slightly reduces the degree of orientation to 0.78 degrees, but pre-melt annealing at 120 °C increases it to 0.83 degrees. In general, the orientation degree can be increased by molding shear, CNTs, and pre-melting annealing.

Without CNTs, the crystallinity of the 0CM can reach 10% due to its long cooling time during the molding process. In contrast, the crystallinity of 0IM is 0, which means that no complete crystal planes can be detected by WAXD. 0IntM slightly increases the crystallinity to 1% by shearing. With the addition of CNTs, the crystallinity of 6IntM quickly increased to 13%. This means that shear alone makes it hard to promote the crystallization of PLA. In contrast, CNTs and the intense extensional flow have synergistic effects on promoting the crystallization of the PLA matrix. Other researchers have also noted this phenomenon, which should be related to the formation of hybrid crystalline fibrils characterized by CNTs coated with PLA extended chains [5]. Afterward, solid-annealing raises the crystallinity to 45%. At the same time, pre-melt annealing, which is above the cold crystallization and prior to the reaching of the melting temperature, can promote cold crystallization, and further expands the crystallinity to 50%.

#### 3.1.2. Crystallinity and Thermal Properties

The DSC curves of heat flow vs. temperature results are shown in Figure 4. Without CNTs and annealing (Curves 1, 2, and 3), the different molding methods almost do not change the melting curves. After adding CNTs (Curves 4, 5, and 6), the cold crystallization peak shifts to a lower temperature, especially for 6IM and 6IntM, which decreases from 101.8 °C to 81.1 °C. This is because CNTs promote the generation of a large number of nucleation sites under the shear field, and shear helps to form pre-order PLA molecular chains, thereby reducing the cold crystallization temperature. In addition, the oriented crystals formed by the CNTs and shear have poor thermal stability, decreasing the melting temperature, as shown in curves 5 and 6. After annealing, the cold crystallization peak disappeared in curves 7, 8, and 9, which means that the annealing at 80 °C can help to finish the crystallization and prevent the occurrence of cold crystallization during the DSC test.

Figure 5a shows the variation in crystallinity and crystal morphology. Without CNTs, the crystallinity of 0IM is only 0.1; 0IntM relies on its higher melt shear strength to form oriented crystals, increasing the crystallinity to 0.15. CM achieves the highest crystallinity of 0.20 by means of a longer cooling time. Without CNTs, shear can only slightly improve the crystallite by about 0.05. At the same time, if CNTs are added, but without shear, and the crystallinity of 6CM is also only 0.05 higher than that of 0CM. However, if the CNTs and shear are involved simultaneously, this will significantly improve the crystallinity, as the crystallinity of 6IntM is 0.15 more than that of 0IM. This means that the synergistic effects of CNTs and shear occurred for the IM and IntM samples, rapidly increasing the crystallinity to 0.3 by forming a hybrid crystal. This synergistic effect also resulted in the crystallinity of IM and IntM samples, exceeding that of CM samples after the addition of CNTs.

As shown in the SEM image in Figure 5(a_1_,a_2_), the CNT agglomerates nucleate during annealing to form a spherocrystal, which can further improve the crystallinity. Through solid annealing at 80 °C for 4 h, the crystallinity of each sample was improved by about 50%. However, pre-melt annealing at 120 °C for 3 h decreased the crystallinity of all samples, possibly due to the fact that a high annealing temperature can melt some low heat resistance crystals.

Figure 5b shows the change in melting point. When CNTs are not added, different melt shear strengths have no significant effect on the melting point. However, with the addition of CNTs, many hybrid crystals are generated, and the melting point is reduced. IntM reduces the melting point to the lowest temperature of 167.5 °C due to its high shear strength. After heat treatment, the melting points of CM, IM, and IntM samples gradually converge. This means that the crystals generated during the annealing have similar structures.

#### 3.1.3. CNT Agglomerates Orientation

As shown in Figure 6, CNTs agglomerate morphology can be observed by OM. For the CIM sample, only a core layer is formed (Figure 6a). CNTs prefer agglomeration due to their high aspect ratio, and the CNTs form surface-to-surface stacking and shoulder-to-shoulder assemblies [31]. CNTs undergo agglomeration and self-assembly during the molding process, forming large structures observed by OM. The CNT aggregates in the CM sample form a network like interconnected structure, which is helpful for the formation of conductive networks.

As shown in Figure 6b,c, long strip shadows can be observed along the flow direction in the shear layer of the IM and IntM samples, which are formed by the alignment of CNT aggregates and dispersed CNTs. This structure contributes to the formation of a unidirectional conductive network. The high shear in IntM cannot orient all CNT aggregates, especially those larger than 300 μm^2^, showing an unoriented state. In addition, the melt shear strength will reach its maximum value near the surface of the mold and gradually decrease as it is away from the mold surface. Therefore, the core layers of IM and IntM Figure 6b’,c’ do not undergo strong shear, resulting in an unoriented structure, which means that the orientation degree of injection molded samples is not uniform. This different shear strength will lead to various particle dispersions in different regions.

Shearing promotes the orientation of CNT aggregates; at the same time, it plays a role in breaking the aggregates. To quantity the dispersion degree of the agglomerates, we introduce the parameter of CNT agglomerate dispersion (*A_dis_*). Due to the fact that the CNTs content of each sample is the same, the number of agglomerates per unit area will increase when the agglomerates break up. Therefore, this paper uses the number of particles per unit area to quantify the *A_dis_*. OM images are used to calculate the agglomerates’ number. Only those agglomerates larger than 10 μm^2^ will be counted in order to avoid errors associated with tiny aggregates. *A_dis_* is equal to the number of agglomerates within 1 mm^2^, and the *A_dis_* can be obtained by Equation (6):(6)Adis=AnumberAarea
where *A_number_* represents the number of agglomerates, and *A_area_* is the OM images area. When an aggregate has been ultimately aggregated into one mass, the dispersion is the worst, and the *A_dis_* equals 1. As the aggregate disperses into particles, *A_dis_* will gradually increase. The *A_dis_* of the shear layer and core layer of CM, IM, and IntM is shown in Figure 7. Without shear, the CM sample gets the lowest *A_dis_*, which is only about 2500 agglomerates in 1 mm^2^. For IM and IntM, the *A_dis_* of the shear layer is much higher than that of the core layer. However, even the core layer of the IM and IntM samples take some flow shear, which lets them get a higher *A_dis_* than that of CM. Although shearing helps disperse aggregates, the highest shearing may not lead to the best dispersion. Because agglomerate distribution is related to many factors, such as particle geometry, agglomeration, shear rate, and viscosity, their influence is hard to predict [32]. In this research, moderate shear stress favors disperse CNT agglomerates; the IM sample then gets the highest *A_dis_* (approximately 32,000) in the shear layer. In general, shear promotes the dispersion of CNTs, making the CNT agglomerates in injection samples more dispersed than that of the compressed ones, and leading to the higher dispersion of aggregates in the shear layer than in the core layer.

### 3.2. Yield and Electrical Properties

#### 3.2.1. Yield Properties

Figure 8a shows the selective stress−strain curves of the samples. 0IntM gets a higher elongation at break than that of 0CM, which is attributed to the lower crystallinity. With the addition of CNTs, the elongation at break decreases, while the modulus and tensile strength increase. After pre-melting annealing, the 6IntM-120 modulus reached its maximum, but its elongation at break significantly decreased, resulting in a 40% decrease in its tensile strength. Figure 8b–d shows the mechanical properties of tensile strength, elongation at break, and modulus separately. Without CNTs and annealing, the yield strength, elongation at break, and elastic modulus of 0IntM were 33%, 42%, and 57% higher than that of 0CM, respectively. This should be due to the formation of oriented structures by high shear stress. Figure 9b shows that 80 °C annealing increases the tensile strength of all samples without CNTs. However, 120 °C annealing reduces the tensile strength, which may be related to the partial degradation of PLA. With the addition of CNTs, the tensile strength of CM samples significantly increased, while the tensile strength of injection molded samples showed little change. Annealing reduces the tensile strength of PLA/CNTs, and the decrease in 6IM and 6IntM is more significant than it is for 6CM. The changing trend in the tensile strength of PLA/CNTs blends is similar to their elongation at break. It can be seen that PLA/CNTs have increased crystallinity in annealing, reduced the number of molecular chains in the amorphous region, and made them more brittle. As a result, the CNTs find it hard to exert their strengthening effect. As shown in Figure 8d, annealing increases the elastic modulus, and the CNTs can improve it further, which is mainly attributed to the increased crystallite. Furthermore, the heat treatment can cause CNTs to rearrange and form a network, reducing the toughness and further enhancing the modulus. Overall, for those PLA/CNTs nanocomposites, both shear and heat treatments are beneficial for improving the tensile modulus, but they deteriorate the tensile strength and elongation at break. Therefore, the tensile modulus of 6IntM120 is 90% higher than that of 0CM, but the tensile strength and elongation at break are only 56% and 27% of the latter.

Figure 9a,c show the tensile fracture surface of 0CM and 6IntM120, respectively. In comparison, the surface of 0CM is very smooth, and some elongated fibers formed by stretching can be observed in its enlarged image (Figure 9b). For 6IntM120, there are no fibers in the fracture surface, but many rough grid structures are formed. From the local magnification photo (Figure 9d), it can be seen that these structures are composed of CNTs. When CNTs create conductive networks, they also increase the crystallite and facilitate the propagation of stress-induced cracks between CNTs networks, thereby accelerating the occurrence of fractures and reducing the elongation at break.

#### 3.2.2. Electrical Properties

The conductivity is detected along the Flow direction (FD) and Transverse direction (TD). The highest activity is achieved by 6CM, while IM and IntM can orient CNTs by injection shear, but with a reduction in their conductivity. This is because the *A_dis_* of the IM and IntM is higher than that of the CM, and higher dispersion discourages the forming of conductive networks. On the other hand, low conductivity can also be attributed to the one-dimensional oriented CNTs in the flow direction [33]. The oriented CNTs bring directional conductivity. As shown in Figure 10b, the CNTs in 6IM are oriented, and the electrical can only translate along the FD. For 6IM, the conductivity difference between FD and TD could be as much as seven orders of magnitude, contributed to by the highest *A_dis_* and CNTs orientation of IM. Oriented CNTs are more likely to connect with each other along the flow direction and form a conductive path, while only a tiny number of unoriented CNTs can complete the conductive path in the TD. In addition, IM also increased the dispersion of CNTs, which further widened the distance between CNTs, resulting in the complete fracture of the pathway in the TD. Thus, the 6 wt.% CNTs is lower than its conductivity threshold in TD. At the same time, although the FD path was partially destroyed by the increased dispersion of CNTs, the main conductive pathway was preserved. As a result, there is a significant gap in resistivity between the FD and TD. However, this gap can be closed by heat treatment. After annealing, the conductivity of both IntM and IM samples rapidly increases along the TD, especially for pre-melting annealing. This is attributed to the rapid increase in crystallinity after a heat treatment, which produces a volume exclusion effect on the CNTs, increasing the local concentration of CNTs and enabling the reconstruction of the CNTs network [34]. In other words, the nucleation effect of CNTs and the crystallized volume exclusion effect coincide. For oriented CNTs, the volume exclusion effect is more significant, transforming the linear conductive network into a three-dimensional conductive network and significantly improving the conductivity in TD. As shown in Figure 10b, crystallization promotes the rearrangement of CNTs, allowing electrons to propagate in multiple directions, and improving the conductivity in the TD. However, for un-oriented CNTs, the volume exclusion effect cannot further enhance the conductivity of the formed three-dimensional conductive network. At this point, the nucleation ability of CNTs and the volume exclusion effect of crystals reach a certain balance, maintaining stable conductivity; for example, the 6CM conductivity is almost unchanged with annealing.

## 4. Conclusions

CM, IM, and IntM provide low, middle, and high shear stress on PLA/CNTs composites, respectively. Without CNTs, or just strong shear or CNTs, it is difficult to improve the crystallinity of PLA. However, shear and CNTs have a synergistic effect, resulting in about a two-fold increase for IM samples. For those PLA/CNTs nanocomposites, both orientation and annealing are beneficial for the improvement of the tensile modulus, but they deteriorate the tensile strength and toughness. In addition, the effect of pre-melting annealing (120 °C) is more pronounced than that of solid annealing (80 °C). Therefore, the tensile modulus of 6IntM120 is 90% higher than that of 0CM, but the tensile strength and elongation at break are only 56% and 27% of the latter.

Proper shear flow can promote the dispersion and orientation of CNTs, resulting in an anisotropic conduction such that that the conductivity difference between FD and TD is as high as seven orders of magnitude. During annealing, the CNTs nucleation and crystallized volume exclusion effects co-occur. For oriented CNTs, the volume exclusion effect is more significant, transforming the linear conductive network into a three-dimensional conductive network and significantly improving the conductivity in TD by seven orders of magnitude, and the anisotropy of conduction is eliminated. For unoriented CNTs, the nucleation ability of CNTs and the volume exclusion effect reach a certain balance, maintaining a stable conductivity.

## Figures and Tables

**Figure 1 materials-16-04012-f001:**
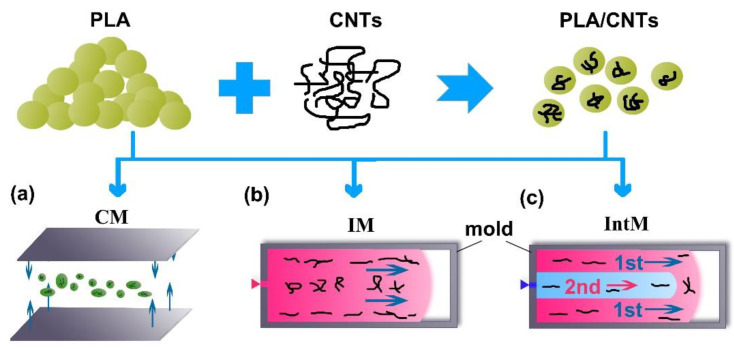
The schematics of the molding methods (**a**) CM, (**b**) IM, (**c**) IntM.

**Figure 2 materials-16-04012-f002:**
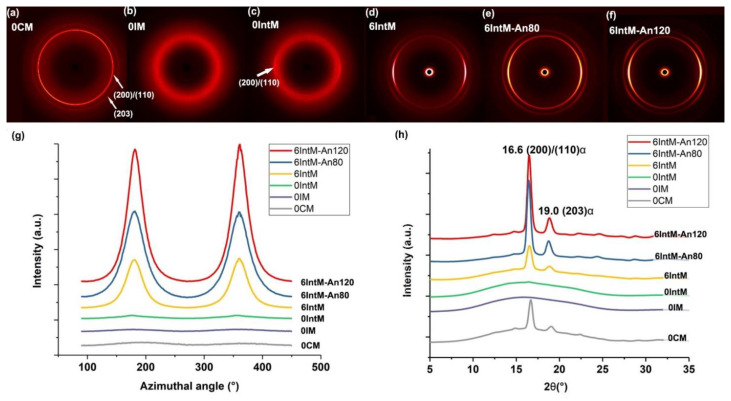
(**a**–**f**) 2D-WAXD patterns of samples, (**g**) azimuthal profiles of (200)/(110) α reflections, (**h**) linear-WAXD curves obtained from circularly integrated intensities of 2D-WAXD patterns.

**Figure 3 materials-16-04012-f003:**
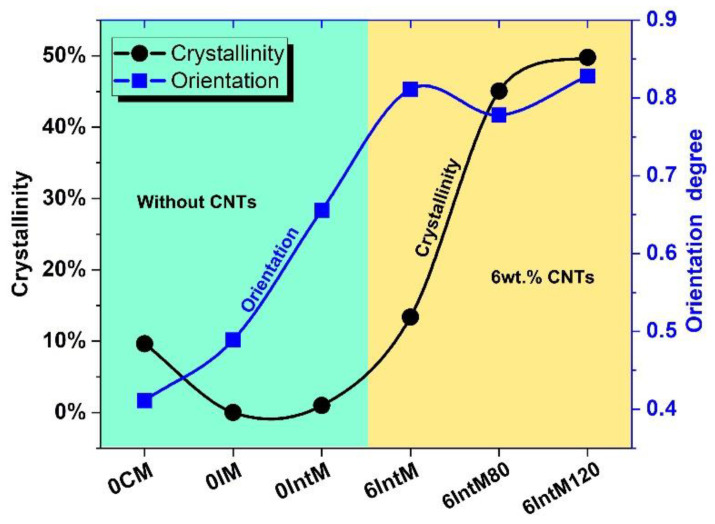
The crystallinity and orientation degree of various samples.

**Figure 4 materials-16-04012-f004:**
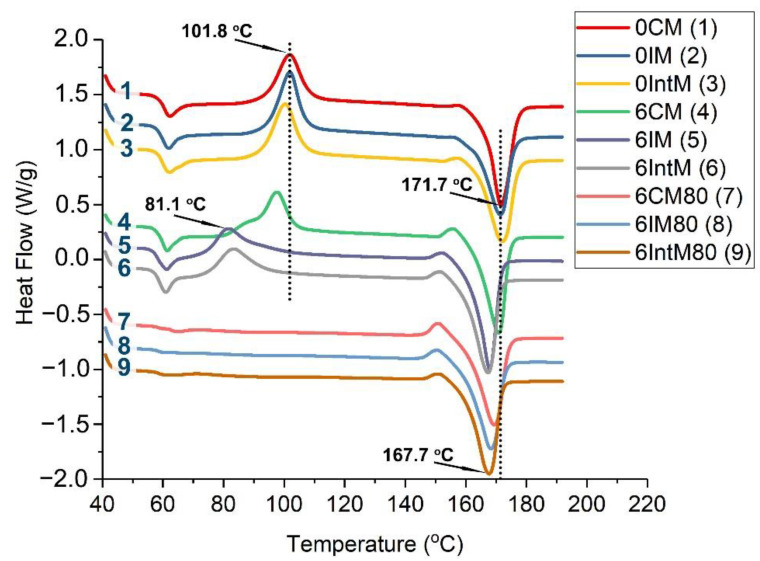
DSC melting curves of various samples.

**Figure 5 materials-16-04012-f005:**
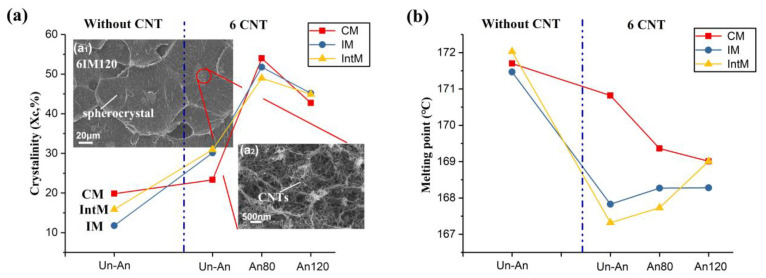
(**a**) the crystallinity and crystal morphology; (**a_1_**) SEM of 6IM120; (**a_2_**) partially enlarged detail; (**b**) the melting point of various samples.

**Figure 6 materials-16-04012-f006:**
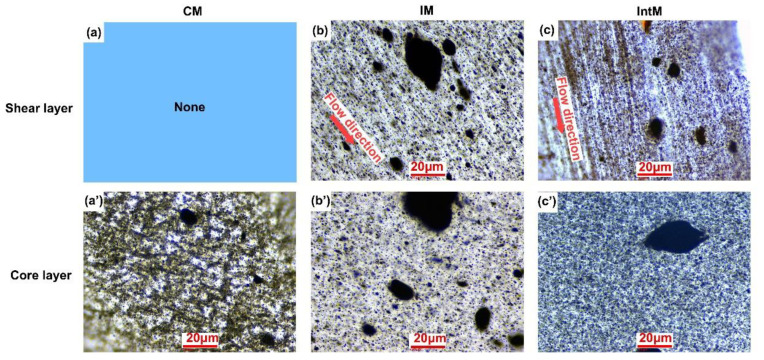
OM micrographs of PLA/CNT nanocomposites; the first row is the shear layer, the second row is the core layer, (**a**,**a’**) 2CM, (**b**,**b’**) 2IM, and (**c**,**c’**) 2IntM.

**Figure 7 materials-16-04012-f007:**
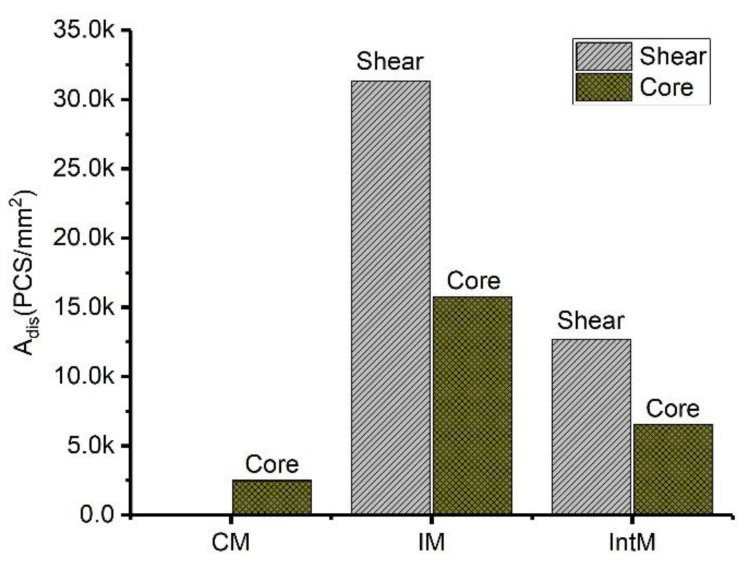
Agglomerate dispersion degree (*A_dis_*) of CM, IM, and IntM samples in the shear and core layers.

**Figure 8 materials-16-04012-f008:**
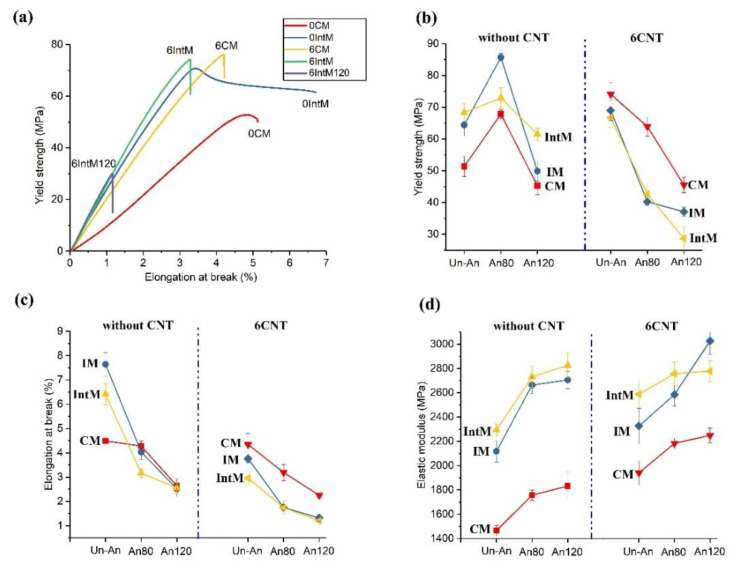
(**a**) Selective stress-strain curves and mechanical properties of various samples; (**b**) the samples’ yield strength; (**c**) the samples’ elongation at break; and (**d**) the samples’ elastic modulus.

**Figure 9 materials-16-04012-f009:**
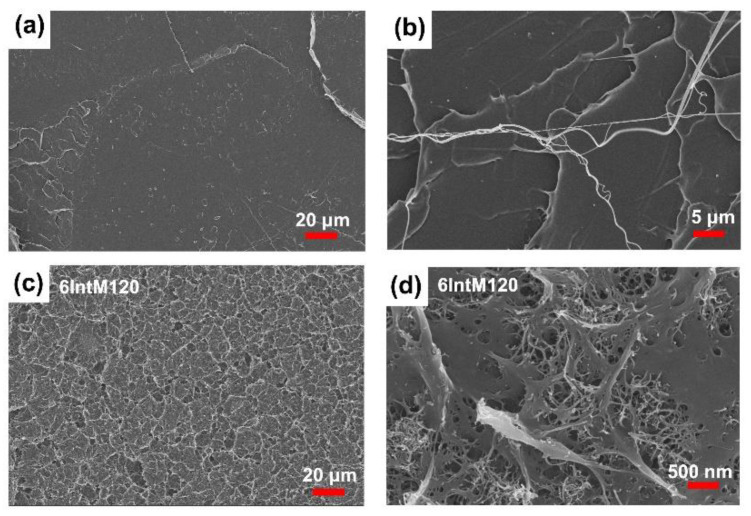
SEM image of the fracture surface of (**a**) 0CM, and (**c**) 6IntM. Local magnification photos are shown in (**b**,**d**).

**Figure 10 materials-16-04012-f010:**
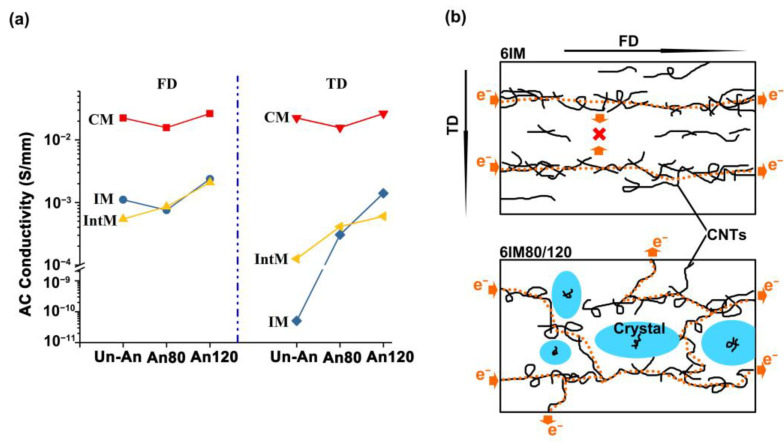
(**a**) Various samples’ direct current conductivity in FD and TD, (**b**) schematic of the conductivity for 6IM and 6IM80/120.

**Table 1 materials-16-04012-t001:** Examples of blend composition and preparation method.

Examples	PLA	CNT	CM/IM/IntM	UN-An/An80/An120
0CM	100 wt.%	0	CM	UN-An
6IntM80	94 wt.%	6 wt.%	IntM	An80

## Data Availability

Some or all data generated or used during the study are available from the corresponding author by request.

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
