# Peer review of "Improved Yield and Electrical Properties of Poly(Lactic Acid)/Carbon Nanotube Composites by Shear and Anneal"

_materials, 2023, doi:10.3390/ma16114012_

Round 1

Reviewer 1 Report

In this study, the authors successfully used three different processing methods to provide three types of shear strength from weak to strong and to investigate the effect of shear and annealing on changes in orientation, crystallinity and CNT distribution of PLA/CNT nanocomposites.

The study is original and contains important contributions to science.

One of the original aspects of this work is the discussion of the nucleation effect and the crystallized volume exclusion effect of CNTs for various oriented CNTs and their ultimate effects on the conductive network and mechanical properties.

The main line of this study was to determine the growth of PLA crystals on the conductive network under different shear conditions by annealing method, followed by the effect of the crystal on the conductive network. The authors have successfully overcome this task.

Although the work is original, some questions should be clarified:

What are the main reasons for anisotropic conductivity where the conductivity difference between FD and TD is as high as 7 orders of magnitude and should be highlighted in the article.

What are the reasons why solid annealing increases crystallinity to 45%, while pre-melting annealing increases crystallinity to 50%?

Why do the core layers of IM and IntM not undergo strong shear and what does the formation of a non-directional structure mean (Fig. 6(b') (c').

What are the main factors that favor the orientation of CNT aggregates and cause clusters to break.

What basic parameters are used in this study to quantify the degree of dispersion of the pellets?

Why is the number of CNT clusters used to represent the distribution?

Overall, the paper is an easy read but is not free from some grammatical mistakes that should be corrected

The article is written well from a technical point of view. The sections are connected with each other very carefully.

The manuscript falls within the scope of the Journal and could be considered for publication after minor revisions.

his study, the authors successfully used three different processing methods to provide three types of shear strength from weak to strong and to investigate the effect of shear and annealing on changes in orientation, crystallinity and CNT distribution of PLA/CNT nanocomposites.

The study is original and contains important contributions to science.

One of the original aspects of this work is the discussion of the nucleation effect and the crystallized volume exclusion effect of CNTs for various oriented CNTs and their ultimate effects on the conductive network and mechanical properties.

The main line of this study was to determine the growth of PLA crystals on the conductive network under different shear conditions by annealing method, followed by the effect of the crystal on the conductive network. The authors have successfully overcome this task.

Although the work is original, some questions should be clarified:

What are the main reasons for anisotropic conductivity where the conductivity difference between FD and TD is as high as 7 orders of magnitude and should be highlighted in the article.

What are the reasons why solid annealing increases crystallinity to 45%, while pre-melting annealing increases crystallinity to 50%?

Why do the core layers of IM and IntM not undergo strong shear and what does the formation of a non-directional structure mean (Fig. 6(b') (c').

What are the main factors that favor the orientation of CNT aggregates and cause clusters to break.

What basic parameters are used in this study to quantify the degree of dispersion of the pellets?

Why is the number of CNT clusters used to represent the distribution?

Overall, the paper is an easy read but is not free from some grammatical mistakes that should be corrected

The article is written well from a technical point of view. The sections are connected with each other very carefully.

The manuscript falls within the scope of the Journal and could be considered for publication after minor revisions

Author Response

Dear Reviewer:

Thank you for your comments. We have studied the comments carefully and have made corrections which we hope meet with approval.

Q: 1. Although the work is original, some questions should be clarified:

What are the main reasons for anisotropic conductivity where the conductivity difference between FD and TD is as high as 7 orders of magnitude and should be highlighted in the article.

A: Thanks for your comments. It is the orientation of CNTs and high dispersion that leads to significant differences in conductivity between FD and TD. This content indeed needs to be emphasized in the text. The article has been modified accordingly.

“For 6IM, the conductivity difference between FD and TD could go as high as 7 orders of magnitude, which contributed by the highest Adis and CNTs orientation of IM. Oriented CNTs are more likely to connect with each other along the flow direction and form a conductive path, while only a small number of unoriented CNTs can complete the conductive path in the TD. In addition, IM also increased the dispersion of CNTs, which further widened the distance between CNTs, resulting in the complete fracture of the pathway in the TD direction. So, the 6wt.% CNTs is lower than its conductivity threshold in TD. At the same time, although the path in FD was also partially destroyed by the increased dispersion of CNTs, the main path was preserved. As a result, there is a large gap in resistivity in the FD and TD.”

Q: 2. What are the reasons why solid annealing increases crystallinity to 45%, while pre-melting annealing increases crystallinity to 50%?

A: Thanks for your question. There should be an explanation here in the text. The manuscript has made corresponding modifications.

“Afterward, solid-annealing raises the crystallinity to 45%, while pre-melt annealing which is above the cold crystallization and before melting temperature, can promote cold crystallization and further expands the crystallinity to 50%.”

Q: 3. Why do the core layers of IM and IntM not undergo strong shear and what does the formation of a non-directional structure mean (Fig. 6(b') (c').

A: Thanks for your comments; the relevant content has been added to explain the molding shear.

“In addition, the melt shear strength will reach its maximum value near the surface of the mold and gradually decrease as it away from the mold surface. Therefore, the core layers of IM and IntM (Figure 6(b ') (c') do not undergo strong shear, resulting in an un-oriented structure, which means that the orientation degree of injection molded samples is not uniform. And this different shear strength will lead to various particle dispersion in different regions.”

Q: 4. What are the main factors that favor the orientation of CNT aggregates and cause clusters to break?

A: The agglomerate dispersion is related to many factors, such as particle geometry, agglomeration, shear rate, and viscosity, and their influence is hard to predict. In this manuscript, the shear should play the most crucial role in the orientation and break of CNT agglomerates. The manuscript has made corresponding modifications.

“In this research, a moderate shear stress favors disperse CNT agglomerates, then the IM sample gets the highest Adis (about 32 thousand) in the shear layer. In general, shear promotes the dispersion of CNTs, making the CNT agglomerates in injection sample more dispersed than that of compressed, and higher dispersion of aggregates in the shear layer than in the core layer.”

Q: 5. What basic parameters are used in this study to quantify the degree of dispersion of the pellets?

A: Thanks for your comments; the basic parameter is the number of CNT aggregates per unit area. More information has been given to explain the agglomerates dispersion (Adis).

“To quantity dispersion degree of agglomerates, we introduce the parameter of CNT agglomerates dispersion (Adis). Due to the CNTs content of each sample being the same, the number of agglomerates per unit area will increase when agglomerates break up. Therefore, this paper uses the number of particles per unit area to quantify the Adis. OM Images are used to calculate the agglomerates’ number.”

Q: 6. Why is the number of CNT clusters used to represent the distribution?

A: Thanks for your question; just as mentioned in the former question: Due to the CNTs content of each sample being the same, the number of agglomerates per unit area will increase when agglomerates break up.

Q: 7. Overall, the paper is an easy read but is not free from some grammatical mistakes that should be corrected.

A: Thanks for your comment. The manuscript has been modified and hope it can reach the standards.

Q: 8. The article is written well from a technical point of view. The sections are connected with each other very carefully.

The manuscript falls within the scope of the Journal and could be considered for publication after minor revisions.

A: Once again, thank you very much for your comments and suggestions.

Reviewer 2 Report

The title needs to be modified.

The authors are suggested to put a table concerning quantity of ingredients in composites.

Quality of Fig. 8 and Fig. 9 must be improved.

The mechanical properties should be discussed more.

Although it is a nice paper but somewhere it lacks in proper discussion.

Needs editing.

Author Response

Dear Reviewer:

Thank you for your comments. We have studied the comments carefully and have made corrections which we hope meet with approval.

Q: The title needs to be modified.

A: Thanks for your comments and the title has been modified.

“Improved Yield and Electrical Properties of Poly(Lactic Acid) / Carbon Nanotube Composites by Shear and Anneal”

Q: The authors are suggested to put a table concerning quantity of ingredients in composites.

A: Thanks for your reminder. this table is necessary.

“Example of the labeled method is also shown in Table 1. ”

Table 1. Examples of blend composition and preparation method.

Examples

PLA

CNT

CM/IM/IntM

UN-An/An80/An120

0CM

100 wt.%

0

CM

UN-An

6IntM80

94 wt.%

6 wt.%

IntM

An80

Q: Quality of Fig. 8 and Fig. 9 must be improved.

A: Thank you for your comments. Figures 8 and 9 have been rearranged to ensure clarity.

Figure 8. (a) selective stress-strain curves, Mechanical properties of various samples, (b) samples’ yield strength, (c) samples’ elongation at break, and (d) samples’ elastic modulus

Figure 9. SEM image of the fracture surface of (a) 0CM, and (c) 6IntM. Local magnification photos are shown in (b) and (d).

Q: The mechanical properties should be discussed more.

A: Thanks for your comments, the discussion about mechanical properties was modified.

“Figure 8(a) shows selective stress−strain curves of samples. 0IntM gets a higher elongation at break than that 0CM, which attribute to the lower crystallinity. With the addition of CNTs, the elongation at break decreases while the modulus and tensile strength increase. After pre-melting annealing, the 6IntM-120 modulus reached its maximum, but its elongation at break significantly decreased, resulting in a 40% decrease in its tensile strength. Figure 8(b), (c), and (d) shows the mechanical properties of tensile strength, elongation at break, and modulus separately. Without CNTs and annealing, the yield strength, elongation at break, and elastic modulus of 0IntM were 33%, 42%, and 57% higher than that of 0CM, respectively. This should be due to the formation of oriented structures by high shear stress. Figure 9(b) shows that 80°C annealing increases the tensile strength of all samples without CNTs. However, 120°C annealing reduces the tensile strength, which may be related to the partial degradation of PLA. With the addition of CNTs, the tensile strength of CM samples significantly increased, while the tensile strength of injection molded samples showed little change. Annealing reduces the tensile strength of PLA/CNTs, and the decrease in 6IM and 6IntM is greater than 6CM. The changing trend in tensile strength of PLA/CNTs blends is similar to their elongation at break. Based on the trend of crystallinity changes, it can be seen that PLA/CNTs have increased crystallinity in annealing, reduced the number of molecular chains in the amorphous region, and made them more brittle. As a result, the CNTs hard to exert their strengthening effect. As shown in Figure 8(d), annealing increases the elastic modulus, and the CNTs can further improve it, mainly attributed to the increased crystallite. Furthermore, the heat treatment can cause CNTs to rearrange and form a network, reducing the toughness and further enhancing the modulus. Overall, for those PLA / CNTs nanocomposites, both shear and heat treatment are beneficial for improving the tensile modulus, but they deteriorate the tensile strength and elongation at break. Therefore, the tensile modulus of 6IntM120 is 90% higher than that of 0CM, but the tensile strength and elongation at break are only 56% and 27% of the latter.”

Q: Although it is a nice paper but somewhere it lacks in proper discussion.

A: Thank you for your approval of the article. We have optimized the discussion section, especially for mechanical and electrical properties.

Q: Comments on the Quality of English Language: Needs editing.

A: Thanks and the entire text has been language modified.

Reviewer 3 Report

 The authors studied the mechanical and thermal effects on PLA/CNT composites. The volume fraction of CNT is important to improve the crystallinity and the orientation of PLA. This is scientifically sounded. In addition, there are enough experimental data. So, this study is worth publishing in this journal. However, I have a comment on the mechanism of crystal orientation, as shown in the following list. If the authors sincerely addressed my concern, this study would meet the criteria for the publication in Materials.

Comment list

Comment 1: Figure 3 showed that the addition of CNT largely enhanced the orientation of crystal. Then, is the orientation of PLA enhanced cooperatively? For example, in the other studies about liquid crystalline blockcopolymer (Nano Lett. 22, 6105 (2022).), the orientation of PEO cylinder is cooperatively changed with that of liquid crystal. By reference to such preceding study, the logic of this paper can be strengthened.

English should be improved partly.

Author Response

Dear Reviewer:

Thank you for your comments. We have studied the comments carefully and have made corrections which we hope meet with approval.

Q: …In addition, there are enough experimental data. So, this study is worth publishing in this journal. However, I have a comment on the mechanism of crystal orientation, as shown in the following list.

Comment 1: Figure 3 showed that the addition of CNT largely enhanced the orientation of crystal. Then, is the orientation of PLA enhanced cooperatively? For example, in the other studies about liquid crystalline blockcopolymer (Nano Lett. 22, 6105 (2022).), the orientation of PEO cylinder is cooperatively changed with that of liquid crystal. By reference to such preceding study, the logic of this paper can be strengthened.

A: Thanks for your suggestion; more discussion and references were added.

“With the increased crystal orientation, the CNTs also get a chance to form one-dimensionally oriented, which may be similar to that of one-dimensionally oriented block copolymer 31.”

  1. Ishibe, T.; Kaneko, T.;  Uematsu, Y.;  Sato-Akaba, H.;  Komura, M.;  Iyoda, T.; Nakamura, Y., Tunable Thermal Switch via Order–Order Transition in Liquid Crystalline Block Copolymer. Nano Letters 2022, 22 (15), 6105-6111.

Round 2

Reviewer 2 Report

Can be accepted 

minor editing required